# Effects of Detritivores on Nutrient Dynamics and Corn Biomass in Mesocosms

**DOI:** 10.3390/insects10120453

**Published:** 2019-12-13

**Authors:** Josephine Lindsey-Robbins, Angélica Vázquez-Ortega, Kevin McCluney, Shannon Pelini

**Affiliations:** 1Department of Biological Sciences, Bowling Green State University, Bowling Green, OH 43403, USA; kmcclun@bgsu.edu (K.M.); spelini@bgsu.edu (S.P.); 2School of Earth, Environment & Society, Bowling Green State University, Bowling Green, OH 43403, USA; avazque@bgsu.edu

**Keywords:** nutrient, macroinvertebrate, eutrophication, phosphorus, nitrogen, carbon, mesocosm, soil health, detritivore, rainfall

## Abstract

(1) Background: Strategies aimed at managing freshwater eutrophication should be based on practices that consider cropland invertebrates, climatic change, and soil nutrient cycling. Specifically, detritivores play a crucial role in the biogeochemical processes of soil through their consumptive and burrowing activities. Here, we evaluated the effectiveness of increasing detritivore abundance as a strategy for nutrient management under varied rainfall. (2) Methods: We manipulated soil macroinvertebrate abundance and rainfall amount in an agricultural mesocosms. We then measured the phosphorus, nitrogen, and carbon levels within the soil, corn, invertebrates, and soil solution. (3) Results: Increasing detritivore abundance in our soil significantly increased corn biomass by 2.49 g (*p* < 0.001), reduced weed growth by 18.2% (*p* < 0.001), and decreased soil solution nitrogen and total organic carbon (*p* < 0.05) and volume by 31.03 mL (*p* < 0.001). Detritivore abundance also displayed a significant interaction effect with rainfall treatment to influence soil total P (*p* = 0.0019), total N (*p* < 0.001), and total C (*p* = 0.0146). (4) Conclusions: Soil detritivores play an important role in soil nutrient cycling and soil health. Incorporating soil macroinvertebrate abundance into management strategies for agricultural soil may increase soil health of agroecosystems, preserve freshwater ecosystems, and protect the valuable services they both provide for humans.

## 1. Introduction

The National Oceanic and Atmospheric Association (NOAA) reported that 2017 was the fourth-worst algal bloom season in history for Lake Erie, Ohio, USA [1]. Long-term studies of lake ecosystems across Europe and North America have indicated that controlling algal blooms and other symptoms of eutrophication depends on reducing inputs phosphorus (P) and nitrogen (N) [2]. Algal proliferation has been shown in response to P additions, to N and P combinations, and the addition of only N [3,4,5,6,7,8]. Large reductions in particulate loads of P and N have been reported in some tributaries of Lake Erie due to the widespread adoption of conservation tillage between 1975 and 1995 that dramatically decreased runoff and erosion [9]. Unfortunately, the more bioavailable, dissolved forms of P and N have increased, causing the stimulation of more toxic strains of algae such as Mycrocystis [10]. Members of this genus cannot convert N_2_ to ammonia (i.e., “fix” N), so they require combined N sources such as ammonium, organic N, or nitrate to support growth. This shift in understanding of harmful algal blooms and their toxicity presents an opportunity for investigation in new nutrient reduction strategies that combine P and N controls.

Freshwater eutrophication management should not rely solely on P and N mitigation, but rather on practices that consider the complexity of ecosystem feedbacks [11]. Macronutrients cycle in soil ecosystems by moving between various pools of nutrients including soil organic matter, soil biota, plants, water-soluble forms in soil solution (i.e., the water and soluble nutrients held within soil that can leach out when soil reaches its water holding capacity) and sorbed to soil particles, as seen in Figure 1. Detritivores, along with other soil biota, contribute to those nutrient pools through death, waste excretion, uptake, mineralization, and immobilization, as seen in Figure 1. The processes of mineralization and immobilization create changing levels of available nutrients to plants as they move between organic and inorganic forms. For example, this includes the incorporation into detritivore biomass or release in their soluble form into soil solution. By way of decomposition, detritivores are able to “unlock” nutrients held within detritus. They can then transport the nutrients into the various stores of nutrients in the soil system, even having indirect effects on plant uptake and the erosion or leaching of nutrients, as seen in Figure 1.

Several studies support that the decomposition and subsequent mineralization of nutrients held in soil organic matter, manure, and plant residue is directly influenced by the consumptive activities of detritivores [12,13,14]. In fact, millipedes and earthworms have been a large focus in soil literature due to their classifications as “ecosystem engineers”, making them a popular study organism for soil health and nutrient dynamics [13,15,16]. Furthermore, ecosystem engineers have been found to play an important role in stimulating the activities of microbial decomposers by increasing substrate availability for microbes by physically processing litter through shredding or by chemically altering litter through digestion [17,18]. Likewise, the chemical and physical properties of detritivore burrows and casts are known to affect microbial community functioning, soil nutrient dynamics [19]. In fact, biopores (i.e., voids in the soil which were formed by the activity of soil life) can change soil hydrology by increasing air transport through the soil, increasing water infiltration, reducing water runoff, and facilitating the acquisition of water and nutrients from the subsoil [20]. Research into soil management, particularly agriculture, should focus on methods that utilize detritivore activities to increase soil health and decrease leaching and runoff into freshwater systems. In 2015, Bender and van der Heijden used a mesocosm study to mimic an agroecosystem, and showed that increased soil macrofauna diversity and abundance improved P mobilization and reduced P leaching by 25% [21]. This suggests that increasing both soil macroinvertebrate diversity and abundance may be useful for increasing nutrient mobilization and immobilization in agroecosystem, thus mitigating nutrient runoff and leachate from agricultural soil.

Changes in precipitation impose numerous threats to ecosystem functioning by possibly altering invertebrate abundance and distribution, nutrient cycling, and plant growth. In a meta-analysis of soil biota responses to climate change, it was found that the abundance of soil fauna decreased with colder or drier conditions [22]. Some species of enchytraeids, a type of segmented worm, alter their vertical distribution with changes in moisture and experience severe mortality under increased temperatures [23]. Shifts in the abundance and distribution of soil organisms can alter their interactions within the soil community, thus changing how they impact their surrounding ecosystem through their consumptive and burrowing activities. Climate change may exacerbate these problems, leading to further impacts on freshwater eutrophication and algal blooms. According to the IPCC, the Midwest could experience a 30% increase in precipitation over the next few decades [24]. The same forecast predicts a 20% increase in rainfall in just the spring (March–May), which coincides with the heaviest fertilization of fields. This increase in precipitation ultimately will alter nutrient cycling by increasing leaching of soluble nutrients. Nutrient losses from agricultural fields are heavily influences by weather-driven fluctuations in leaching rather than changes in agricultural production or management practices [25].

The purpose of this study was to evaluate the effectiveness of increasing invertebrate abundance as a strategy for nutrient management in an agroecosystem. We aimed to answer three questions: (1) How does increased soil invertebrate abundance influence the need for fertilizer to optimize crop production? We hypothesized that soil invertebrates would enhance P or N availability in soil, therefore leading to higher P and N uptake in crops. Thus, we predicted that higher macrofauna abundance would increase plant biomass beyond that resulting from fertilizer use only. (2) How does increased soil macroinvertebrate abundance influence P and N mobilization and leaching? We hypothesized that, because soil invertebrates would increase P and N mobilization, less P and N would be lost from the system via leaching. (3) How does soil macroinvertebrate abundance interact with precipitation changes to influence P and N mobilization and leaching? We hypothesized that, if macrofauna increase mobilization, then precipitation increases would not affect P or N leaching in mesocosms with increased soil macrofauna abundance. We predicted to observe a “buffering” effect, where high invertebrate abundance decreased P and N leaching, even in mesocosms within the 20% increased precipitation treatment.

## 2. Materials and Methods

### 2.1. Study System

We established mesocosms in the BGSU greenhouse in Bowling Green, OH. The greenhouse is not climate controlled so temperature and humidity were controlled using windows, vents, and fans while being monitored using iButtons (Maxim Integrated, San Jose, CA, USA). The soil used in this experiment was collected from an agricultural field in Lucas County, Ohio and is classified as mixed, mesic aquatic Udipasamments of the Ottokee series (92.6% sand, 1.3% silt, 6.1% clay). The agricultural field at which the soil was collected has a history of corn and soy bean rotation, application of anhydrous ammonia, pot ash, lime, and liquid ammonium polyphosphate (10-34-0), and conventional tillage.

### 2.2. Experimental Design

A total of 30 mesocosms were constructed and subject to one of two precipitation treatments: historical mean or elevated 20% above mean according to the U.S. Global Change Research Program projections for northern Ohio [24]. This resulted in 15 mesocomsm per precipitation experiment. We altered the abundance of earthworms (*Lumbricus terrestris*), millipedes (*Narceus americanus*), and pill bugs (*Armadillidium* spp.) within each mesocosm while keeping species evenness the same. Within each precipitation treatment, there were five invertebrate abundance levels, including a control of zero. Each abundance level increased the number of individuals of each species by 1, so there was between 0 and 4 individuals of each species in the mesocosm. Each combination of invertebrate abundance and precipitation level was replicated three times. We re-evaluated the invertebrate abundance of each mesocosm at the end of the experiment to determine final species abundance and mortality.

### 2.3. Mesocosm Construction and Maintenance

Mesocosms were constructed using 5 gallon plastic buckets (height 29.2 cm, diameter at top 30.2 cm, diameter at bottom 26.2 cm). A circular hole 6 cm in diameter was cut into the center of the bottom of each bucket and covered with 2 mm aluminum window screen to allow for adequate drainage and to prevent organisms from escaping. Plastic funnels were attached underneath each hole and fixed with clean, removable plastic bottles that were used to collect soil solution (Appendix A). The buckets were placed in the BGSU greenhouse and suspended off of the ground between wooden planks. Before the start of the experimental trial, the outside rim of each mesocosm was brushed with Tanglefoot sticky trap (Tanglefoot, Marysville, OH, USA) to prevent outside invertebrates from crawling into the mesocosms, while the inside rim was brushed with Insect-a-slip (BioQuip, Rancho Dominguez, CA, USA) to keep invertebrates inside the mesocosm. We did not observe any millipedes, pill bugs, or earthworms trapped in the Tanglefoot sticky trap or present outside of the mesocosms, thus the escape of experimental invertebrates is assumed to not have affected the final invertebrate abundances. While we did observe a number of flies in the Tanglefoot sticky trap, no other invertebrates were observed, indicating that outside pests or predators did not enter the mesocosms.

### 2.4. Soil Collection and Preparation

Soil was collected on April 2, 2018 after litter and surface detritus was raked away. Due to tillage, we did not encounter soil layering, thus the soil was not separated by depth when reconstructed in the mesocosms. The soil was sieved through 0.5 cm wire mesh to remove macroinvertebrates, roots, rocks, and large pieces of detritus and to maintain the naturally occurring soil aggregates. We homogenized the soil using the cone and quarter method which involved (1) piling soil onto a plastic tarp forming a cone shape, (2) raking quartered sections of the pile towards four opposing directions, and (3) shoveling the distributed soil around to other quarters to evenly disperse the soil before reforming the original cone [26]. The cone and quarter method was applied to the soil three times to ensure adequate mixing, and the soil was visually inspected for residual invertebrates before being added to the mesocosms. During homogenization and soil sieving, we observed a handful of earthworms, beetle larvae, and crab spiders, but they were removed before putting the soil in the mesocosms.

We added a total of 13 liters of soil to each bucket (20 cm in depth). After the soil was transferred to the mesocosms, three seeds of organic field corn (Reid’s Dent Corn (*Zea* spp.)) were planted. After germination, we thinned the corn to only one plant per mesocosm. Over the course of the experiment, we observed other small plants grow in the mesocosms, which we classified as “weeds” because they were not the intended crop. We counted each weed individually at the end of the experiment prior to mesocosm destruction to obtain weed abundance. We applied 58 grams of 0.02N-0.02P-0.02K Fertilizer (Miracle-Gro, Marysville, OH, USA) three times during weeks 3, 9, and 11 due to observations of nitrogen limitation in the corn including the yellowing and death of lower corn leaves.

To allow for the stabilization and acclimation of microbial communities and their functioning, the mesocosms remained undisturbed for four weeks after set up, but prior to the start of the experiment (weeks 1–4), except for a daily administration of 100 mL ultrapure water to prevent the death of microbial communities and the corn. We measured mesocosm soil moisture daily with a soil moisture probe (Delta-T Devices SM150, Cambridge, UK). Throughout weeks 1–4, it was noted that a level of 15%–30% soil moisture was adequate for corn growth and did not exceed soil water holding capacity, thus soil moisture was maintained at 15%–30% throughout the experimental trial (weeks 6–13) through daily additions of ultrapure water as needed.

### 2.5. Synthetic Rainwater and Storm Design

Ultrapure water was collected in sterile 20 L carboys and adjusted to mimic rainwater pH and electrical conductivity (hereafter “synthetic rainwater”). The electrical conductivity was adjusted to 78 µS/cm using approximately 0.7 g NaCl upon collection and the pH was adjusted to 5.2 daily using 20 M HCl. A total of 5 man-made ‘storms’ occurred throughout the experimental trial during which synthetic rainwater was added to each mesocosm using a horticultural watering can to simulate a rainfall event equivalent to a 25.4 mm storm for 45 min. The average rainfall treatment received 1332 mL of synthetic rainwater for this simulation, while the elevated rainfall treatment received 20% more synthetic rainwater at 1599 mL. These storms were in addition to the regular watering for plant water balance.

### 2.6. Invertebrate Assemblages

Earthworms (*Lumbricus terrestris*), millipedes (*Narceus americanus*), and pill bugs (*Armadillidium* spp.) were purchased from Carolina Biological Supply (Burlington, NC, USA). Invertebrates were placed in plastic deli containers with moist paper towels for 24 h allowing them to clear their guts. The healthy individuals (based on physical appearance and activity level) were weighed and added to the mesocosms at the beginning of week 5. Precipitation treatments began one week after invertebrates were added to the mesocosms (week 6) to allow for adequate acclimation. Invertebrate casting, molting, and mortality was recorded throughout the duration of the experiment as visualized on the surface of the mesocosms.

### 2.7. Soil Solution Nutrient Analysis

Soil solution was collected immediately following the storms to minimize evaporation and 24 h following each storm, for a total of five separate collection dates spaced 7–10 days apart. Samples of soil solution were filtered into sterile Whirl Pak B01062 sampling bags (Cincinnati, OH, USA) and refrigerated immediately after collection in order to be analyzed for ortho-phosphates, ammonium, nitrate/nitrite, total N, and total P using a SEAL AQ2 discrete analyzer (Seal Analytics, Mequon, WI, USA) [27] (Appendix B). Dissolved total C was determined using high temperature oxidation followed by infrared detection of CO_2_ using a Shimadzu TOC-VCSH equipped with a liquid auto sampler Shimadzu ASI-L (Shimadzu, Kyoto, Japan) (Appendix C).

### 2.8. Mesocosm Harvest

Mesocosms were harvested haphazardly over a two-day period during week 13 due to signs of heat-stress in the invertebrates and nitrogen limitation in the corn. We observed widespread death of millipedes as well as excessive burrowing of pill bugs. The corn was displaying yellowing and drying on basal leaves, indicating nitrogen and water limitation. On the first day of harvest, after weeds were counted, we cut each corn plant at the base of the stalk and placed it into a paper bag. Additionally, we took two soils cores with a diameter of 2 cm at a depth of 10 cm from the soil surface; one core was extracted from the center of the mesocosm, and another core was taken from around the circumference of the mesocosm. If detritivore burrows were visible at the soil surface, the second core was targeted on those areas. This targeting is necessary because chemical and physical properties of detritivore burrows and casts are known to affect microbial community functioning and soil nutrient dynamics [19]. We placed the soil cores for each mesocosm in separate plastic bags, and they were homogenized for 1 min and air-dried at room temperature for 23 days.

On the second day of harvest, we destructively sampled the mesocosms to remove surviving invertebrates. First, we collected all visible organisms at the surface and placed them in plastic containers with moist paper towels. Second, each mesocosm was dumped onto a tarp for below-ground organism and corn root collection. We gently washed the corn roots with water to remove soil and placed them in paper bags for oven drying. Once all surviving invertebrates were found, they were kept in a plastic container with a moist paper towel for 24 h to clear their gut, and biomass was recorded 24 h later. We recorded mortality during this time as the number of invertebrates not found. Even if whole carcasses of individuals were found, they were not counted as part of ending invertebrate abundance. Corn stalks, leaves, roots, ears, and invertebrates were dried in an oven at 60 °C until a constant mass was reached.

### 2.9. Soil Nutrient Analysis

Soil samples from each mesocosm were analyzed for total P, total N, and total C at the end of the drying period. Each soil sample was ground using a mortar and pestle and sieved to maintain homogenous particle size for the analytical machines (0.841 mm, No. 20 mesh). Total P and total N were analyzed using SEAL AQ2 discrete analyzer (Seal Analytics, Mequon, WI, USA) after an acid-potassium persulfate digestion [27] (Appendix B). Soil samples were analyzed for total C using a Shimadzu TOC-L/SSM-5000A (Shimadzu, Kyoto, Japan) (Appendix D).

### 2.10. Corn and Invertebrate Nutrient Analysis

Dried corn leaf samples were ground using a mortar and pestle (0.420 mm, No. 40 mesh) before undergoing potassium persulfate digestion and colorimetric analysis of total N and total P using a SEAL AQ2 discrete analyzer (Seal Analytics, Mequon, WI, USA) (Appendix B) [28,29]. Dried invertebrates were pooled by species into one sample from each mesocosm, ground in a mortar and pestle (0.420 mm, No. 40 mesh), and analyzed for total N and total P using a SEAL AQ2 discrete analyzer (Seal Analytics, Mequon, WI, USA) (Appendix B). Apple Leaves (NIST^®^ SRM^®^ 1515) were used as a certified reference material for a standard measure of P and N in living organisms. Dried corn leaves and invertebrates were also analyzed for total C using a Shimadzu TOC-L/SSM-5000A (Shimadzu, Kyoto, Japan). Laboratory grade Dextrose (S25295A, Fisher Science Education, Nazareth, PA, USA) was used as a standard for living organisms during total C analysis of invertebrates (and BBOT Leco Certified Reference Material was used as a standard during total C analysis of corn leaves (Appendix D).

### 2.11. Statistical Analysis

All analyses were performed using R (R Core Team, 2016). We used the glm package to fit general linear models (GLMs), analyzing the main effects and interaction effects between macroinvertebrate abundance and precipitation treatment for corn biomass, invertebrate mortality, soil nutrients, corn nutrients, and invertebrate nutrients. Due to repeated measures following storm events, we used the nlme package to fit Linear Mixed-Effects Models (LMEs), analyzing the main effects and interaction effects between invertebrate abundance and precipitation treatment for soil solution volume and soil solution nutrients (total P, total N, total C, ortho-phosphates, nitrate/nitrite, and ammonia). We included mesocosm ID as a random variable and considered multiple temporal autocorrelation structures (compound symmetry, autoregressive, unstructured) by including the date of each leachate collection. A continuous AR (1) correlation was determined to be the best temporal autocorrelation structure, thus the corCAR1 function was included in LMEs [30]. Unfortunately, Kenward–Roger approximation has not been implemented for the nlme package, thus it could not be used on the LMEs. Due to this, our ANOVA results for LMEs, based on Wald’s tests, may be somewhat anticonservative, i.e., *p*-values somewhat close to 0.05 may be suspect, and not indicative of a real biological effect [31]. A two-way ANOVA using the car package was used to analyze each model to test for the main effects of macroinvertebrate abundance and precipitation treatment and the interaction of each treatment. Least-squares means and least-squares trends in the lsmeans package were used for post-hoc analyses to compare linear combinations, estimate slopes of trend lines, and estimate treatment group means of GLMs and LMEs when significant relationships were found. In all cases, statistical significance was accepted at α ≤ 0.05. Assumptions of normality and equal variance were checked via visual examination of plots of residuals. Measures of soil solution volume were normalized using a square root transformation.

## 3. Results

### 3.1. Invertebrate Mortality and Nutrients

Ending invertebrate abundance was significantly affected by starting invertebrate abundance (*p* < 0.001, χ^2^ = 54.5), as seen in Table 1. Each intended initial treatment level of invertebrate abundance increased final invertebrate abundance by 22%, showing that distinct abundance levels existed despite invertebrate mortality throughout the experiment. At the end of the experiment, no earthworms were found in any treatment.

Millipede and pill bug body nutrient samples were pooled among replicates in invertebrate treatment groups for each rainfall treatment to obtain sufficient sampling material. Due to this limitation in sample size (n = 4), the interactive effects of invertebrate abundance and rainfall on invertebrate nutrient content were not analyzed. We did not detect significant additive relationships between millipede total P and rainfall treatment (*p* = 0.345, χ^2^ = 0.9) or invertebrate abundance (*p* = 0.901, χ^2^ = 0.02), as seen in Table 1. Millipede total N decreased significantly as invertebrate abundance increased (*p* = 0.0284, χ^2^ = 4.8), but was not significantly affected by rainfall treatment (*p* = 0.638, χ^2^ = 0.22). Total N within millipedes decreased by approximately 0.49% for each invertebrate added. Millipede total C also did not show a significant relationship with rainfall treatment (*p* = 0.342, χ^2^ = 0.9) or invertebrate abundance (*p* = 0.338, χ^2^ = 0.9).

Pill bug total P was significantly increased under elevated rainfall when compared to historical rainfall (*p* = 0.00379, χ^2^ = 8.37). Pill bugs under the elevated rainfall treatment had a mean of 0.0124 mg P, while pill bugs under historical rainfall treatment had a mean of 0.0104 mg. Pill bug total P significantly increased with invertebrate abundance (*p* = 0.0341, χ^2^ = 4.49), but we note that due to our use of anticonservative Wald’s tests, this result should be viewed with caution [31]. For every invertebrate added, there was an increase of 0.01% in pill bug total P within elevated rainfall, but only a 0.09% increase in pill bug total P within historical rainfall. We did not detect a significant relationship between pill bug total N and rainfall treatment (*p* = 0.297, χ^2^ = 1.1), as seen in Table 1, or invertebrate abundance (*p* = 0.598, χ^2^ = 0.3). While we did not detect a significant effect of rainfall on pill bug total C (*p* = 0.412, χ^2^ = 0.7), we did find that pill bug total C concentration for each mesocosm increased significantly with invertebrate abundance, with a 66% increase in total C for each pill bug added (*p* = 0.0405, χ^2^ = 4.2), but we note that due to our use of anticonservative Wald’s tests, this result should be viewed with caution.

### 3.2. Soil Nutrients and Ratios

Soil total P displayed a significant interaction effect between rainfall and invertebrate abundance (*p* = 0.0019, χ^2^ = 9.6), as seen in Figure 2a. Soil total P response to invertebrate abundance was contingent upon rainfall treatment, with total P in the historical rainfall treatment decreasing significantly as invertebrates increased, but the trend was insignificant for the elevated rainfall treatment. Similarly, we observed a significant interaction effect between rainfall and invertebrate abundance for soil total N (*p* = 0.000382, χ^2^ = 12.6), as seen in Figure 2b. Within historical rainfall, soil total N decreased as invertebrate abundance increased, but the opposite was observed for soil under elevated rainfall. Soil total C also displayed a significant interaction effect between rainfall and invertebrate abundance (*p* = 0.0146, χ^2^ = 6.0), as seen in Figure 2c. Within elevated rainfall, soil total C increased significantly with increasing invertebrate abundance, but historical rainfall did not display a significant trend. Soil C:P ratio displayed a significant direct relationship with invertebrate abundance, with soil C:P increasing approximately 1.2% for every invertebrate added (*p* = 0.0169, χ^2^ = 65.7), as seen in Figure 3a. However, soil C:P was not significantly altered by rainfall treatment (*p* = 0.368, χ^2^ = 0.8), as seen in Table 1. Under elevated rainfall, the C:P ratio was 0.651, whereas the mean C:P ratio under historical rainfall was only slightly lower at 0.527. Similarly, soil C:N ratio displayed a significant direct relationship with invertebrate abundance, with soil C:N increasing approximately 0.31% for every invertebrate added (*p* = 0.218, χ^2^ = 5.3), as seen in Figure 3b. Soil C:N did not change significantly with rainfall treatment (*p* = 0.427, χ^2^ = 0.6). Within elevated rainfall, the mean soil C:N ratio was 0.173, while the soil C:N ratio under historical rainfall was slightly lower at 0.148. There was no significant difference detected between the soil N:P ratios between rainfall treatments (*p* = 0.967, χ^2^ = 0.002), as seen in Table 1, or invertebrate abundance levels (*p* = 0.923, χ^2^ = 0.009), as seen in Table 1.

### 3.3. Corn Biomass and Nutrients

Corn total mass (including roots, stalk, fruit, and leaves) displayed a significant direct relationship with invertebrate abundance, increasing by approximately 2.49 g with every invertebrate added (*p* < 0.001, χ^2^ = 24.3), as seen in Figure 4. However, we did not observe a significant relationship between corn total mass and rainfall treatment (*p* = 0.412, χ^2^ = 0.6). There was a significant increase in corn aboveground biomass (stalk, leaves, and fruit) with invertebrate treatment (*p* < 0.001, χ^2^
*=* 21.9), but not with rainfall treatment (*p* = 0.436, χ^2^ = 0.6). Corn belowground biomass (roots) also displayed a significant positive direct relationship with invertebrate abundance (*p* < 0.001, χ^2^ = 13.0), but did not respond significantly to rainfall treatment (*p* = 0.468, χ^2^ = 0.5). Corn aboveground mass increased approximately 1.97 g with every invertebrate added while corn belowground mass increased approximately 0.05 g with every invertebrate added. Despite only 14 out of the 30 mesocosms producing fruit due to lack of time, corn ear mass was also weighed at the conclusion of the experiment. We did not detect a significant response of corn ear mass to rainfall treatment (*p* = 0.256, χ^2^ = 1.3), as seen in Table 1, or invertebrate treatment (*p* = 0.204, χ^2^ = 1.6), as seen in Table 1.

We did not observe significant differences in corn total P concentration (*p* = 0.896, χ^2^ = 0.5), total N concentration (*p* = 0.909, χ^2^ = 0.01), or total C concentration (*p* = 0.357, χ^2^ = 0.8) between rainfall treatments, as seen in Table 1. Similarly, invertebrate treatment did not significantly impact corn concentrations of total P (*p* = 0.469, χ^2^ = 0.5), total N (*p* = 0.426, χ^2^ = 0.6), or total C (*p* = 0.90, χ^2^ = 0.01). After mass balance calculations, invertebrate abundance was found to significantly increase corn total P (*p* < 0.001, χ^2^ = 16.1), as seen in Figure 5a, total N (*p* = 0.0025, χ^2^ = 9.1), as seen in Figure 5b, and total organic C (*p* < 0.001, χ^2^ = 14.7), as seen in Figure 5c, by 3.5 mg, 19.7 mg, and 0.00095 mg per invertebrate added, respectively. Rainfall treatment did not significantly change corn total P (*p* = 0.528, χ^2^ = 0.34), total N (*p* = 0.637, χ^2^ = 0.223), or total organic C (*p* = 0.746, χ^2^ = 0.10), as seen in Table 1.

### 3.4. Weed Abundance

The total number of weeds (any noncorn plant) had a significant inverse relationship with invertebrate abundance (*p* < 0.001, χ^2^ = 186.6), as seen in Figure 6. There was an 18.2% reduction in weeds for each invertebrate added. However, weed abundance did not display a significant interaction with rainfall treatment (*p* = 0.654, χ^2^ = 0.2), as seen in Table 1.

### 3.5. Soil Solution Volume and Nutrients

The volume of soil solution decreased significantly as invertebrate abundance increased when pooled across all storm dates (*p* < 0.001, χ^2^ = 20.9), as seen in Figure 7. While soil solution decreased approximately 31.03 mL for every invertebrate that was added, we did not detect a significant relationship between soil solution volume and rainfall treatment (*p* = 0.0624, χ^2^ = 3.4). We did not detect a significant effect of invertebrate abundance on soil solution total C (*p* = 0.641, χ^2^ = 0.2), total N (*p* = 0.559, χ^2^ = 0.3), total P (*p* = 0.241, χ^2^ = 1.4), orthophosphates (*p* = 0.194, χ^2^ = 1.7), nitrate/nitrite (*p* = 0.398, χ^2^ = 0.7), or ammonia (*p* = 0.983, χ^2^ = 0.0004), as seen in Table 1. Similarly, rainfall treatment did not significantly alter soil solution total C (*p* = 0.251, χ^2^ = 1.3), total N (*p* = 0.715, χ^2^ = 0.1), total P (*p* = 0.642, χ^2^ = 0.2), orthophosphates (*p* = 0.569, χ^2^ = 0.3), nitrate/nitrite (*p* = 0.251, χ^2^ = 0.8), or ammonia (*p* = 0.769, χ^2^ = 0.09), as seen in Table 1.

Mass balance calculations were completed for each storm by multiplying soil solution nutrient concentrations by volume for each mesocosm. We did not detect a significant difference in total P (*p* = 0.953, χ^2^ = 0.004), ammonia (*p* = 0.286, χ^2^ = 1.13), orthophosphates (*p* = 0.763, χ^2^ = 0.09), or nitrate/nitrite due to invertebrate abundance (*p* = 0.897, χ^2^ = 0.017), as seen in Table 1. However, invertebrate abundance significantly decreased total N (*p* = 0.007, χ^2^ = 7.25), as seen in Figure 8a, and total organic C (*p* = 0.04, χ^2^ = 4.18), as seen in Figure 8b, by 0.816 mg and 0.46 mg, respectively, but we note that due to our use of anticonservative Wald’s tests, this result should be viewed with caution. Rainfall had no detectable effect on total N (*p* = 0.053, χ^2^ = 3.7), total P (*p* = 0.379, χ^2^ = 0.77), ammonia (*p* = 0.1, χ^2^ = 2.7), orthophosphates (*p* = 0.672, χ^2^ = 0.18), nitrate/nitrite (*p* = 0.730, χ^2^ = 0.12), or total organic C (*p* = 0.32, χ^2^ = 0.97), as seen in Table 1.

## 4. Discussion

The results of this study demonstrate that detritivores substantially contribute to agriculture’s ecological impact by influencing nutrient use efficiency. Overall, we found that increasing detritivore abundance in the soil significantly increased corn biomass (*p* < 0.001), as seen in Figure 4, reduced weed growth (*p* < 0.001), as seen in Figure 6, and decreased soil solution volume (*p* < 0.001), as seen in Figure 7. It also decreased total organic carbon and nitrogen (*p* < 0.05), as seen in Figure 8, in soil solution after mass-balance calculations. Depending on rainfall treatment, detritivore abundance also significantly influenced soil total P (*p* = 0.0019), total N (*p* < 0.001), and total C (*p* = 0.0146), as seen in Figure 2. These results support our hypothesis that detritivores increase soil nutrients through their role in decomposition, and buffer the changes in precipitation caused by climate change by decreasing soil solution volume and nutrients. Nevertheless, the mechanisms behind these observations need further testing. Overall, this study showed that soil detritivores play critical roles in nutrient cycling and soil health, but their influence is contingent on rainfall.

### 4.1. Total Soil Nutrients

We found that soil total P was reduced as invertebrate abundance increased under historical low rainfall. The trend is reversed under elevated rainfall. This suggests that under higher rainfall conditions, detritivores increase the amount of total P within the soil, possibly leading to higher bioavailable P in soil. Soil total N also increased with invertebrates under higher rainfall, creating the potential for increased bioavailable pools of N for plants. The observed contingency of soil nutrient levels on precipitation could have several explanations.

First, past studies have found that soil detritivore feeding activity and isopod-driven decomposition are highly contingent on soil moisture and rainfall frequency [32,33]. The detritivores under elevated rainfall may have increased consumption or excrement creation, thus increasing soil nutrients. We observed that soil C:P and C:N ratios increased significantly with detritivore abundance despite rainfall amount. Detritivores are important soil engineers and play a critical role in decomposition by shredding detritus, which leads to the release of key nutrients that are trapped within plant tissues. The results suggest that the millipedes and pill bugs in the mesocosms contributed to the total C in the soil, thus increasing C:P and C:N ratios, possibly through the decomposition of the corn residue and other detritus in the mesocosms. In fact, the results for soil total C support this idea, showing that soil total C increased significantly with invertebrate abundance under elevated rainfall. However, this increase in total C in conjunction with detritivore abundance was not present under historical rainfall. This indicates a contingency of invertebrate activity on rainfall amount which has been documented in other studies.

Second, soil nutrient levels may be linked to rainfall due to plant exudation of organic acids in relation to evapotranspiration. Studies have shown that plant roots continually respond to and alter their immediate environment through the function and regulation of root exudates [34,35]. The complicated relationships observed between rainfall treatment and soil nutrients may be a result of root exudation of organic acids. Root exudation is highly dependent on soil moisture and plant water requirements. Under elevated rainfall, plants may have increased water uptake due to the extra soil moisture, thus exuding more nutrients in the process. Further research is needed to expand upon the relationship between soil moisture, soil nutrients, and root exudation in an agroecosystem dominated by corn. Our observations of changes in soil total P and total N include pools and forms of those nutrients, such as water-soluble orthophosphates, nitrate/nitrite, and phosphate-bearing minerals. Therefore, future research is needed to analyze these pools of P and N separately to elucidate the impact of these changes in soil nutrient composition have for plants. Nevertheless, the results imply that detritivores have the potential to improve the pools of total P and total N within soil.

Additionally, studies have linked increases in microbial biomass, a secondary measure of microbial activity, to increases in the C:P and C:N ratios due to further microbial immobilization of soil C [36,37]. The removal of detritivores and other consumers in detrital food webs from heterotrophic decomposition systems has been shown to decrease the activity of soil microbes dramatically, leading to reduced N and C mineralization [38]. Microbial biomass has been shown to be negatively related to soil solution P, presenting the opportunity to utilize microbial immobilization of nutrients as a management strategy to reduce P in soil solution [37]. Future studies should examine the specific effects detritivore abundance has on soil total C, microbial biomass, and mineralized forms of P, N, and C in soil solution in order to establish possible mechanisms by which detritivores are impacting the nutrient cycle in soil.

### 4.2. Soil Solution Nutrients

Throughout the five artificial storms that occurred in the experiment, higher levels of detritivore abundance reduced the amount of soil solution that percolated through our cropping system. Detritivores reduced the amount of leached soil solution by 31.03 mL for every individual detritivore added, a substantial number considering the small scale of this experiment. This reduction in percolated soil solution may be explained by an increase in corn biomass that leads to increased evapotranspiration. Detritivores may have reduced leachate by increasing evapotranspiration from greater corn biomass. This increased evapotranspiration could lead to drier soils, which could absorb more water during storms, leading to less leached soil solution.

Moreover, this change in leached soil solution volume significantly reduced the load of total N and total organic C that was lost from the mesocosms. Not only did detritivores significantly reduce the amount of water that left the soil, but they also reduced the overall load of nutrients that were carried away with the soil solution after storms. This supports that increased soil invertebrate abundance could be used as a management strategy to reduce the amount of runoff and leachate from agroecosystems. This reduction in soil solution volume, total N, and possibly total organic C was even observed in mesocosms under 20% elevated rainfall, indicating that enhanced detritivore abundance in agricultural soil may be able to buffer leaching during extreme rainfall events that the Midwest is predicted to experience under climate change.

### 4.3. Weed Abundance

Enhanced detritivore abundance reduced weed growth, which indicates that soil macroinvertebrates may enhance nutrient-use efficiency in our cropping system. We found a reduction of weed abundance by approximately 18.2% for every individual detritivore added to the mesocosms, as seen in Figure 5. This reduction in weed growth may reflect altered feeding preference by the detritivores. Millipedes, pill bugs, and other detritivores have been shown to primarily consume leaf litter, wood, dead plant roots, and other dead plant matter [39,40,41]. Selective feeding by detritivores has been extensively studied and is considered to be mediated by litter traits such as nutrients, lignocellulose content, and colonization of microorganisms [42,43,44,45,46,47]. Under conditions with limited resources, such as in agricultural soil with low detritus, we predict that detritivores could alter their feeding preferences and consume smaller plants or seeds. Such a change in foraging preference could account for the decline we observed in the abundance of smaller weeds in the mesocosm. However, we did not directly observe feeding behavior during this experiment. In fact, scientists are looking into integrated weed management programs that utilize “weed seed predators” such as crickets to act as biological control agents to control weed populations in agricultural systems [48].

Another possible explanation for the observed reduction in weed abundance is a plant response to the defensive compounds secreted by millipedes. Millipedes can release a wide array of compounds that are highly repellent to most vertebrate and invertebrate natural enemies, with the potential to also harm plants. Members of at least eight genera of millipedes have been shown to release toxic compounds, including Rhinocricus, Spirobolus, Spirostreptus, Iulus, and Polyceroconas [49]. The species used in this study, *N. americanus*, belongs to the Spirobolidae family, and has long been studied for its ability to release toxic compounds when threatened [50]. The exact compounds released are highly specific to genera; however, most of the compounds for Spirobolidae have been classified as benzoquinones and are effective at killing or deterring mites, fungi, and bacteria [51]. The power of benzoquinones to deter microbes and other microbiota lies in its ability to prolong the lag phase of microbial growth. During this phase, they cause a disruption in the reduction ability of the cells (i.e., the ability of an organism to carry out oxidation-reduction reactions) [52]. It is possible that these compounds interacted with the weeds within the mesocosm, leading to interrupted plant growth, providing further explanation for the reduced weed abundance in mesocosms. We were unable to determine the exact mechanism by which millipedes and pill bugs reduced weed growth. Our results indicate that increasing detritivore abundance in agricultural soil has a negative impact on weed abundance. Furthermore, by reducing the number of weeds growing in the mesocosm, millipedes and pill bugs decreased competition between weeds and corn for soil nutrients and water, which may preface higher nutrient and water uptake by the intended crop.

### 4.4. Corn Biomass

Corn biomass was increased by approximately 2.49 g for every invertebrate added to the system, which may be linked to higher nutrient uptake in the corn due to decreased weed abundance. One of the most important factors influencing plant biomass is soil nutrient availability. Ecological stoichiometry predicts that plant growth rate is characterized by a specific ratio of RNA to protein and this ratio has been linked with the organisms’ N:P ratio [53]. The N:P ratio within plant tissue is highly dependent on N and P levels in the environment. Studies that examine the effect of N addition on plant biomass are relatively abundant and have found that N addition generally increases plant N:P ratio [54,55,56,57,58,59]. Plants can also alter biomass allocation to below- or aboveground plant structures in times of nutrient limitation. Increases in underground biomass allocation has been shown in response to deficiency of both N and P, but the effect of N is usually stronger [60,61]. Alternatively, plants with a high N:P ratios normally allocate less biomass to roots than plants with low N:P ratios [62,63]. Due to the measured increase in corn total biomass with enhanced invertebrate abundance, we can speculate that the invertebrates likely enhanced soil nutrient availability. This enhancement then allows the corn to fulfill its N and P needs and allocate those resources to biomass production. Our study did not find any significant relationships between detritivore abundance and nutrient concentrations within the corn, but we did find an increase in overall corn nutrients after mass-balance calculations. This can be expected if increased soil nutrients related to detritivore activity led to increased corn biomass, keeping the per gram nutrient content of the corn the same, while increasing the total amount of nutrients in the plant as a whole.

### 4.5. Future Work and Implications for Agriculture Management Strategies

This project was designed to expand our knowledge on the role of detritivore abundance in nutrient cycling, and its role in the global freshwater eutrophication crisis. Soil organisms are an integral component of ecosystems, but little recognition is given to their activities and role in agricultural systems. Our study found that higher detritivore abundance decreased weed abundance, increased corn biomass, and decreased soil solution volume. Further work in this field should specifically test whether nutrient-use efficiency is higher in agricultural field sites with increased soil biota, particularly macroinvertebrates and detritivores. By testing similar variables in actual fields, we may be able to get a better idea of how to incorporate detritivore abundance into best management strategies (BMPs). A large fraction of nutrients in applied fertilizer react quickly with the soil environment, rendering it unavailable to plants, and causing the overapplication of fertilizer [64]. Our results demonstrate that detritivores can help achieve higher crop biomass, reduced nutrient loss in soil solution, and increased soil organic matter. Increasing detritivore activity could reduce the need for globally limited nutrient resources, leading to more sustainable agricultural practices.

Likewise, future research should examine how cover crops, conservation tillage, crop rotation, and other agricultural management strategies may influence the colonization of fields with macroinvertebrates. It has been shown that no-tillage practices can increase soil total C, microbial biomass, and N and C mineralization over conventional tillage practices [65]. Additionally, no-till or conservation tillage provide conducive environments for both soil fauna by providing soil cover for food and habitat and regulating soil moisture and temperature [66]. Future work should begin to incorporate measures of detritivore abundance and diversity into each analysis of BMPs to begin to assess the viability of incorporating detritivores into agricultural ecosystems. By incorporating soil life and soil health standards into existing BMPs, we may be able to have even better regulation of the nutrient losses from fields and improve the overall sustainability of agriculture. This study has served as one of the first steps identifying the potential for agricultural soil invertebrates to help preserve freshwater ecosystems and protect the valuable services they provide for humans.

## 5. Conclusions

We conclude that detritivores significantly improve soil health by contributing to nutrient immobilization, mineralization, and mobilization. Through their waste excretion, consumptive activities, burrowing, and even their death, they help to move nutrients between pools within soil. In our agricultural study system with enhanced detritivore abundance, we found increased corn biomass, reduced weed growth, and decreased soil solution volume. Depending on rainfall treatment, detritivore abundance also significantly influenced soil total P, total N, and total C. Our prediction that detritivore abundance would buffer nutrient leaching under elevated rainfall was also supported, as overall nitrogen and total organic carbon were decreased in soil solution across all storm events. Most importantly, our study revealed the mountain of possibility that lies within enhancing soil health through the introduction of soil macroinvertebrates. Existing BMPs such as cover crops, conservation tillage, and crop rotation increase habitat quality and food sources for detritivores, enticing them to colonize agricultural systems. Future studies should creatively test new BMPs to determine the role that soil macroinvertebrates may play in creating more efficient and sustainable agriculture.

## Figures and Tables

**Figure 1 insects-10-00453-f001:**
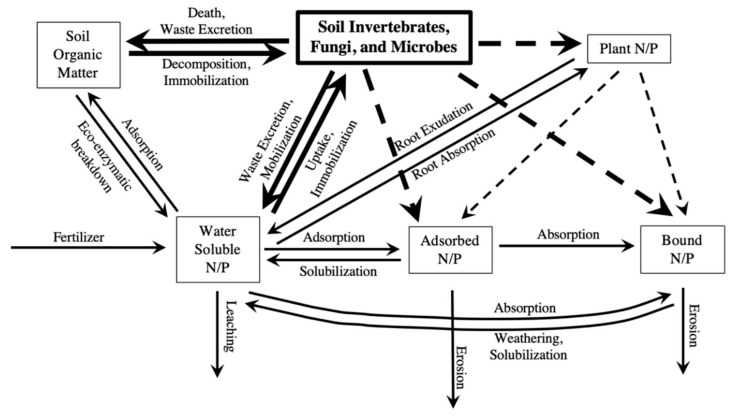
Diagram of basic pools of P and N in soil. Arrows indicate specific mechanisms that contribute to the movement and cycling of those nutrients between pools, entering the soil ecosystem, and leaving the soil ecosystem. Dotted arrows indicate indirect effects between pools that may occur as a result of the direct effects.

**Figure 2 insects-10-00453-f002:**
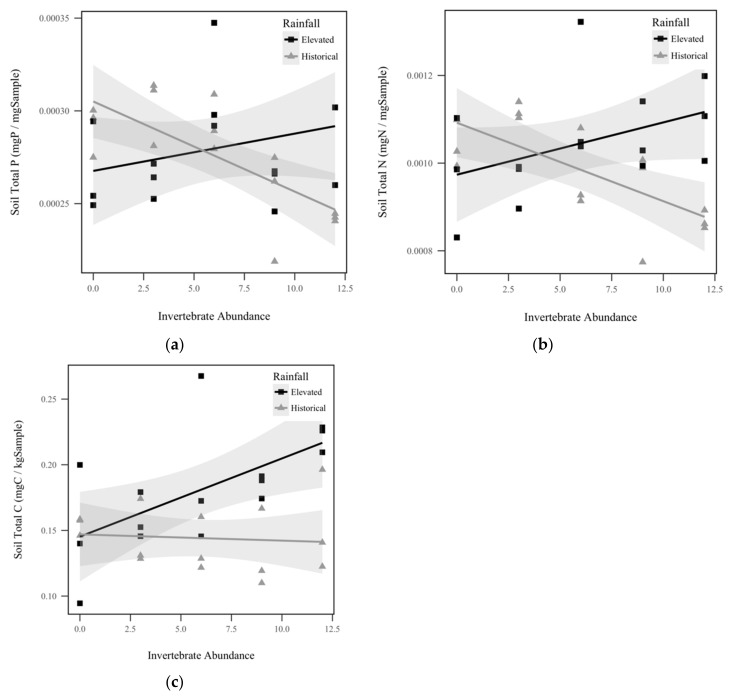
Soil nutrients by historical rainfall (gray triangles) and elevated rainfall (black squares) treatments: (**a**) Soil total P (mg P/mg Sample) displayed a significant interaction between invertebrate abundance and rainfall treatment (*p* = 0.0019, χ^2^ = 9.6); (**b**) Soil total N (mg N/mg Sample) displayed a significant interaction between invertebrate abundance and rainfall treatment (*p* = 0.000382, χ^2^ = 12.6); (**c**) Soil total C (mg C/kg Sample) displayed a significant interaction between invertebrate abundance and rainfall treatment (*p* = 0.0146, χ^2^ = 6.0). Shadowed regions indicate the 95% confidence interval.

**Figure 3 insects-10-00453-f003:**
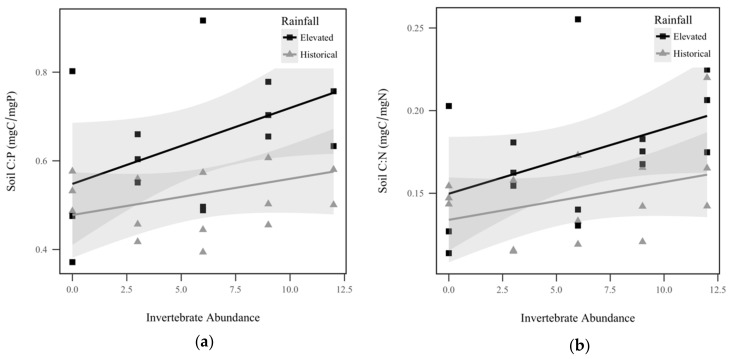
Soil nutrient ratios by historical rainfall (gray triangle) and elevated rainfall (black square) treatments: (**a**) Soil C:P (mg:mg) was significantly correlated with invertebrate abundance (*p* = 0.0169, χ^2^ = 65.7); (**b**) Soil C:N ratio (mg:mg) was significantly correlated with invertebrate abundance (*p* = 0.218, χ^2^ = 5.3). Shadowed regions indicate the 95% confidence interval.

**Figure 4 insects-10-00453-f004:**
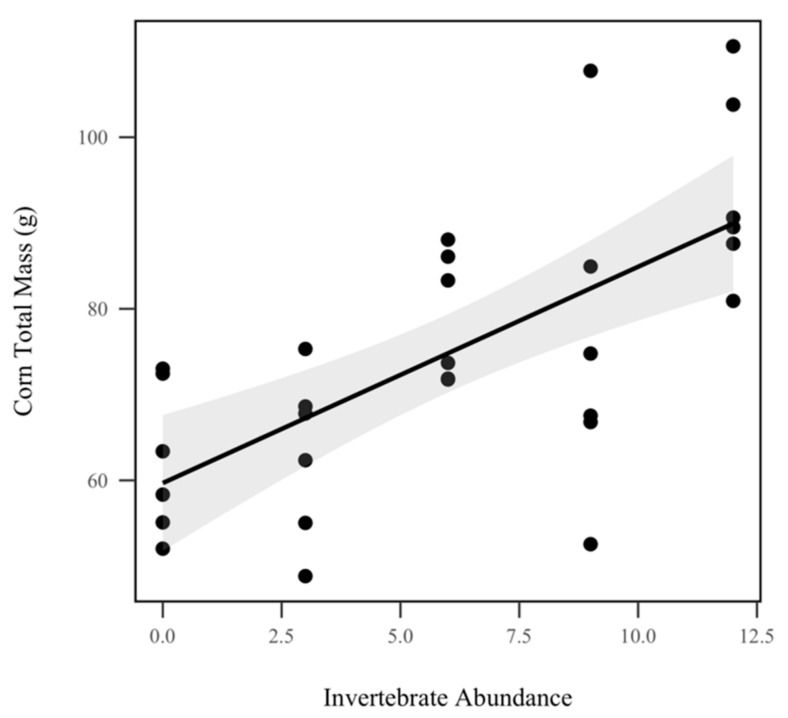
Corn total biomass (g). Invertebrate abundance was significantly correlated with corn total biomass (*p* < 0.001, χ^2^ = 24.3). Shadowed regions indicate the 95% confidence interval.

**Figure 5 insects-10-00453-f005:**
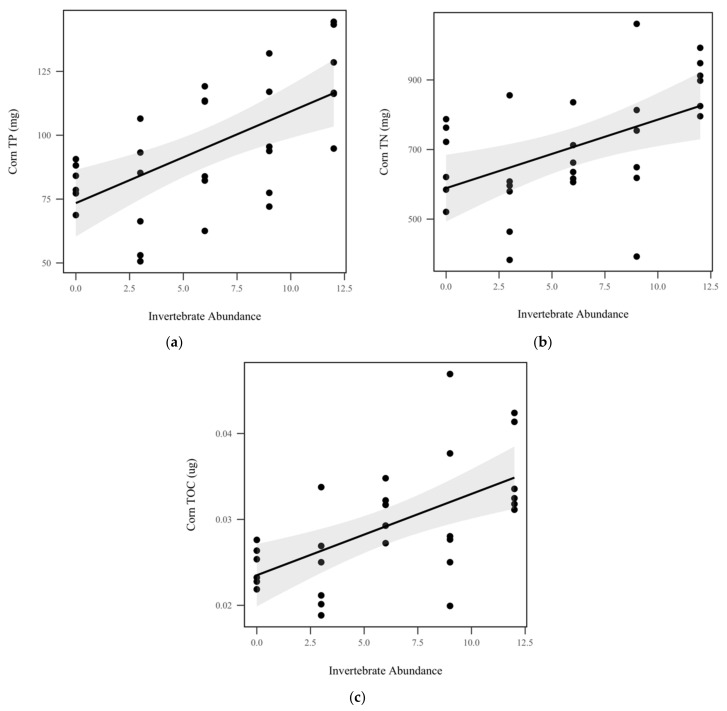
Mass balance of corn nutrients: (**a**) Corn total P (TP) (mg) significantly increased with invertebrate abundance (*p* < 0.001, χ^2^ = 16.1); (**b**) total N (TN) (mg) significantly increased with invertebrate abundance (*p* = 0.0025, χ^2^ = 9.1); (**c**) total organic C (TOC) (µg) significantly increased with invertebrate abundance (*p* < 0.001, χ^2^ = 14.7). Shadowed regions indicate the 95% confidence interval.

**Figure 6 insects-10-00453-f006:**
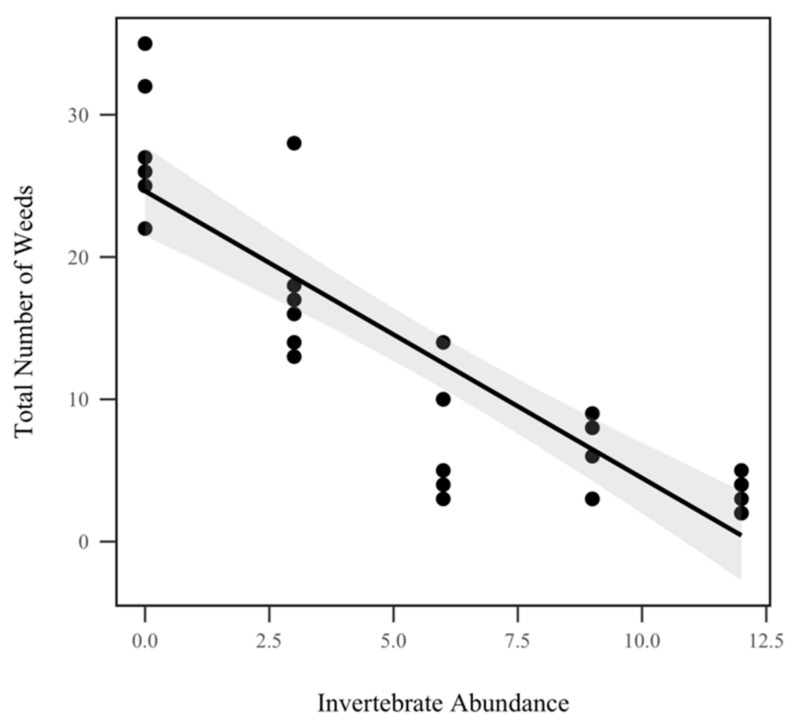
Total number of weeds in each mesocosm by invertebrate abundance. Total weed abundance was significantly correlated with invertebrate abundance (*p* < 0.001, χ^2^ = 186.6), but not with rainfall. The shadowed region indicates the 95% confidence interval.

**Figure 7 insects-10-00453-f007:**
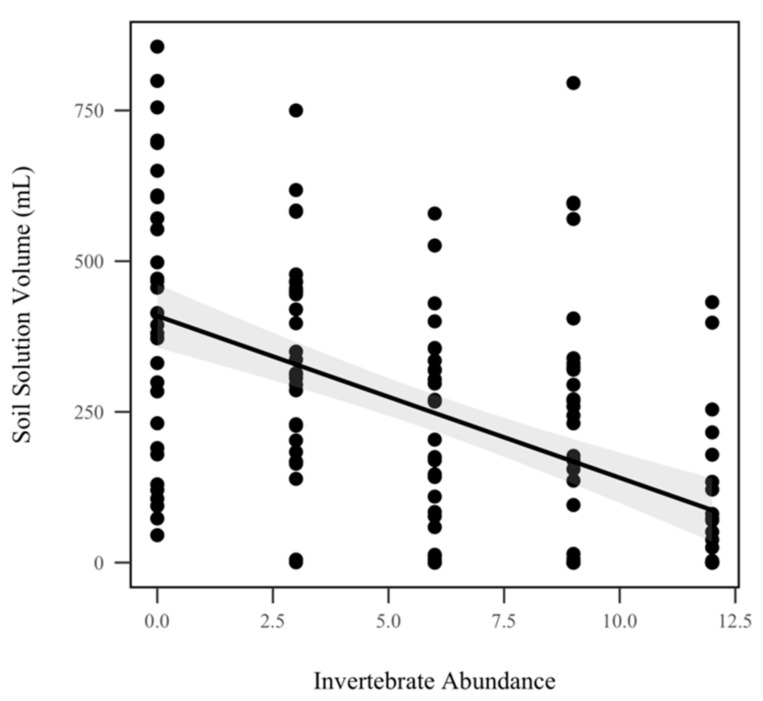
Soil solution volume (mL) for all five storms. Soil solution volume was significantly correlated with invertebrate abundance when pooled across all storm dates (*p* < 0.001, χ^2^ = 20.9). The shadowed region indicates the 95% confidence interval.

**Figure 8 insects-10-00453-f008:**
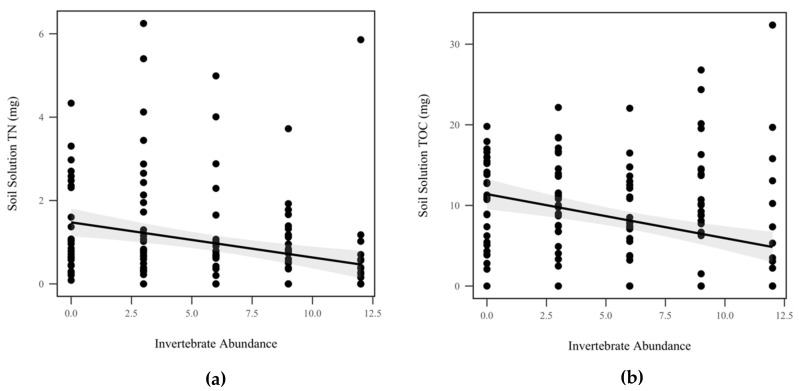
Mass balance of soil solution nutrients: (**a**) total N (TN) (mg) decreased significantly as invertebrate abundance increased (*p* = 0.007, χ^2^ = 7.25); (**b**) total organic C (TOC) (mg) decreased significantly as invertebrate abundance increased (*p* = 0.04, χ^2^ = 4.18). Shadowed regions indicate the 95% confidence interval.

**Table 1 insects-10-00453-t001:** Summary of chi-square values. Invertebrate abundance and rainfall interaction effects were not tested for millipede and pill bug variables due to a lack of statistical power and limited sample size.

		χ^2^ of Predictor Variables		
Response Variables		Invertebrate Abundance	Rainfall Treatment	Invertebrate Abundance and Rainfall Interaction
Mesocosm n = 30	Ending Invertebrate Abundance	54.5 *	0	1.1
	Total Weed Abundance	186.6 *	0.2	1.5
Corn n = 30	Aboveground Biomass	21.9 *	0.6	0.03
	Belowground Biomass	13.0 *	0.5	0.09
	Corn Ear Biomass	1.6	1.3	0.03
	Total Biomass	24.3 *	0.6	0.1
Corn Nutrient Concentrations n = 30	Total P	0.5	0.02	0.4
	Total N	0.6	0.01	0.02
	Total C	−0.01	0.8	0.06
Corn Mass Balance Nutrients n = 30	Total P	16.1 *	0.3	0.4
	Total N	9.1 *	0.2	0.0004
	Total C	14.7 *	0.1	0.03
Soil n = 30	Total P	1.7	5.3 *	9.6 *
	Total N	0.5	3.7	12.6 *
	Total C	4.3 *	0.008	6.0 *
	C:P	5.7 *	0.8	0.7
	C:N	5.3 *	0.6	0.4
	N:P	0.009	0.002	0.01
Millipedes n = 4	Total P	0.02	0.9	-
	Total N	4.8 *	0.22	-
	Total C	0.9	0.9	-
Pill bugs n = 4	Total P	4.49 *^,1^	8.37	-
	Total N	0.3	1.1	-
	Total C	4.2 *^,1^	0.7	-
Soil Solution Nutrient Concentrations n = 30	Total P	1.4	0.2	1.2
	Total N	0.3	0.1	2.5
	Total Organic C	0.2	1.3	2.6
	Total NH^4+^	0.0004	0.09	0.7
	Total PO_4_^3−^	1.7	0.3	1.5
	Total NO^3+^/NO^2−^	0.7	0.8	1.4
	Volume	20.9 *	3.4	0.006
Soil Solution Mass Balance Nutrients n = 30	Total N	7.3 *	3.7	0.02
	Total P	0.004	0.8	0.07
	Total Organic C	4.2 *^,1^	0.9	0.3
	Total NH^4+^	1.1	2.7	0.6
	Total PO_4_^3−^	0.09	0.2	0.05
	Total NO^3+^/NO^2−^	0.02	0.1	0.1

^1.^ We note that due to our use of anticonservative Wald’s tests, this relationship should be viewed with caution and may not represent a true effect [31]. * Significant values denoted with an asterisk (*p* < 0.05).

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
