# Peer review of "Effects of Detritivores on Nutrient Dynamics and Corn Biomass in Mesocosms"

_insects, 2019, doi:10.3390/insects10120453_

Round 1
Reviewer 1 Report
Comments for Authors
Josephine et al. tested the effect of detritivores abundance and rainfall on nutrient dynamics and corn biomass in an agricultural mesocosms. They found that higher detritivore abundance increased corn biomass, reduced weed growth, and decreased soil solution nutrients. This study provided the information of nutrient concentrations in soil, soil solution, corn and detritivores under different detritivores abundance. However, there is no analysis about nutrient dynamics, it’s only relationship between each of them with detritivores abundance or rainfall. Besides, this study was conducted in an agricultural mesocosms, and there was only one plant in each mesocosm, it’s hard to connect this study with field study, and there were no control plots such as mesocosms without plant.
Another problem is the English language, some sentences were too long, it really difficult to understand. And please also check the commas and so on. The authors should make a thorough check on writing to improve the general readability of the paper.
Special comments:
Title: When I first saw the title, I thought it was conducted in a field (agroecosystems), can you adjust it?
Introduction:
The aim of this study was to evaluate increasing detritivores abundance for nutrient management under varied rainfall. Algal bloom was the results of nutrient leaching such as N and P. However, I don't think it's necessary to talk about this problem for the first two paragraphs, can you delete some sentences and combine them to one?
Line 53: HABs, what is shorten for?
Line 106: Three questions and their hypothesis overlapped in some point such as decrease leaching. The last hypothesis was not answering your question about interact with precipitation. I suggested that you reorganize these questions and hypothesis.
M&M:
Line 141: Can you provide a sketch map of the mesocosm?
Line 197: How about abundance? You mentioned five levels, how to adjust it?
Results:
Line 281: Please added table number at least once for each paragraph.
From figure 5 to 8, you didn’t separate historical rainfall and elevated rainfall.
Discussion:
Line 435-438: The sentence was too long, and the commas was misplaced. There were many other sentences like this, please check it through!
Line 444: Missing point.
Line 464: impact of
Line 548-549: why? If corn biomass increased 2.49g for every invertebrate.
Figure 4: There was no black square. And shadowed region was not shown unless I click the figure. It was the same for figure 5,6,7,8.
Reviewer 2 Report
This manuscript describes an experiment that tested the effects of increased rainfall and detritivore abundance on crop yield and nutrient dynamics in mesocosms. The presentation of the study was very straightforward and the results should be of interest to researchers of cropland ecosystems, global change, and aquatic pollution. This was a very thorough study and I found only one problem with the manuscript. On page 374, lines 371-372, the legend for Figure 4 notes that historical rainfall is represented by gray triangles and elevated rainfall is represented by black squares. However, neither the triangles or squares appear in the figure. Overall, this appears to be a well written manuscript describing a very solid study with interesting results.
Reviewer 3 Report
comments by line:
105-118: one aspect could have been added; the physical part of soil-hydrology: the soil invertebrates leave in the soil vertical connected macropores or biopores, which have internally even a bioactive lining. Soil water from excessive rainfall can drain in these pores, without surface runoff (erosion, pollution) and without “leaching” the soil matrix. This is an additional factor contributing to the expected effects. Another factor is, that soil aggregates formed by invertebrate action in the soil are more stable in water than soil aggregates from non organic sources.
569-577: conventional tillage is a hostile environment for soil invertebrates. It destroys their habitat and it exposes with bare soil surfaces the soil to extreme temperatures. The thoughts developed in this direction regarding no-till and its effects on soil invertebrates should be extended a little bit more. This study is excellent, as it explains, why Conservation Agriculture (no-till+soil cover+crop diversity) produces less nutrient contamination in ground and surface water. There are many studies available on the soil biology of Conservation Agriculture systems (for example by Jill Clapperton or Marie Bartz or by the Manitoba-North Dakota no-till organization MANDAK), but not many of these studies describe the actual effects analyzed in this study. However, they confirm, that Conservation Agriculture with its combination of no-till for a stable habitat and soil cover for feed, temperature and moisture regulation of the soil provide conducive environments for soil meso and macrofauna. It would be good to mention this in this paragraph or the conclusions.
Round 2
Reviewer 1 Report
Well done.
Author Response
Thank you so much! We appreciate your helpful comments and your role in making our manuscript stronger.